# Toxicological Response of Zebrafish Exposed to Cocktails of Polymeric Materials and Valproic Acid

Alexandra Savuca [1,2], Ionut-Alexandru Chelaru [1,2,*], Ioana-Miruna Balmus [2,3], Alexandrina-Stefania Curpan [2], Mircea Nicusor Nicoara [1,4,*] and Alin Stelian Ciobica [4,5,6,7]

1. Doctoral School of Geosciences, Faculty of Geography and Geology, "Alexandru Ioan Cuza" University of Iași, Carol I Avenue, 20A, 700505 Iași, Romania; alexandra.savuca@yahoo.com
2. Doctoral School of Biology, Faculty of Biology, "Alexandru Ioan Cuza" University of Iași, Carol I Avenue, 20A, 700505 Iași, Romania; balmus.ioanamiruna@yahoo.com (I.-M.B.); andracurpan@yahoo.com (A.-S.C.)
3. Department of Exact Sciences and Natural Sciences, Institute of Interdisciplinary Research, "Alexandru Ioan Cuza" University of Iasi, Alexandru Lapusneanu Street, 26, 700057 Iasi, Romania
4. Department of Biology, Faculty of Biology, "Alexandru Ioan Cuza" University of Iași, Carol I Avenue, 20A, 700505 Iași, Romania; alin.ciobica@uaic.ro
5. Center of Biomedical Research, Romanian Academy, Iasi Branch, Teodor Codrescu 2, 700481 Iasi, Romania
6. Academy of Romanian Scientists, 3 Ilfov, 050044 Bucharest, Romania
7. Preclinical Department, Apollonia University, 700511 Iasi, Romania
* Correspondence: chelaru.alexandru@yahoo.com (I.-A.C.); mirmag@uaic.ro (M.N.N.)

**Abstract:** Microplastic pollution represents an emerging problem of great interest in the public domain in the last decade; in addition, it overlaps with another delicate problem—pollution with pharmaceutical products that can have negative effects on the environment and people, even in small amounts. The main purpose of this study was to assess the biochemical and behavioral effects of exposure of adult zebrafish (*Danio rerio*) to polyethylene (PE), polypropylene (PP) and valproic acid (VPA), respectively to their mixtures—possible situations in natural aquatic environments. In terms of behavioral responses, sociability appears to be more impaired in the PP group after 5 days of exposure. The mechanisms affected are more those of swimming performance than of sociability. Even more, VPA increases presence in the arm with conspecifics but decreases mobility and locomotion, indicating a possible anxiety mechanism. The mixtures decrease the aggressiveness, especially in the case of the PE+VPA group, where it reaches a super low level compared to the control, which could endanger the species in nature. Regarding the anxiogenic effect, PP and PE act differently: if PE has an anxiogenic effect, on the opposite side is the PP group, which shows a bolder and more agitated behavior. All four variants showed behavioral changes indicative of toxicity from the first dose.

**Keywords:** microplastics; valproic acid; zebrafish; toxicity assessment

## 1. Introduction

Plastic pollution has been growing problem in recent years, gaining great public interest [1]. Both terrestrial and aquatic environments are affected by the increase in polymer pollution. Larger plastics gradually degrade into mesoplastics (5–20 mm diameter) and microplastics (<5 mm), which are difficult to degrade naturally [2]. This is an alarming problem for fresh water and sea water as well. Polyethylene and polypropylene are some of the most common plastics found in rivers and oceans. The two types of polymers have applications in both the construction and packaging industries [3].

An additional sensitive issue is pharmaceutical pollution, which may adversely affect the environment and people [4]. Even in small amounts, these pharmaceutical wastes can have a detrimental effect on the environment [5]. Therefore, recent research has focused on the elimination of these contaminants, using biotransformation methods on the microbial

communities treating contaminated water [6]. Nowadays, pharmaceutical residues have started to be monitored due to their presence in wastewater treatment plant effluents and surface waters [7]. A current concern is that some of the active pharmaceutical ingredients partially persist in wastewater treatment processes [8]. The presence of these pharmaceutical substances is due to consumers through the use and disposal of medicines [9]. The use of wastewater for irrigation results in increased pharmaceutical pollution [10].

Valproic acid is the tolerated [11], prescribed medicine for the treatment of epilepsy [12,13], as a mood stabilizer [14,15]. It is also known as a pediatric hepatotoxic agent [16]. VPA is also known to be prescribed to treat affective disorders, spinal muscular atrophy, and headache [17]. Exposure to various environmental chemicals, both in air and in water, has been shown to be a trigger or even a cause of certain neurological disorders, for example, autism spectrum disorder [18]. Valproic acid is one of the pharmaceutical substances presenting an environmental risk in Iraq [19], also reported as high environmental risk in Switzerland since 2010 [20]. Some studies have shown that samples from the influent of the Back River, USA, and from the Baltimore, MD, wastewater treatment plant contained 130 ng $L^{-1}$ [21] and 140 ng $L^{-1}$ of valproic acid [22], respectively [21,22].

The animal model zebrafish has been gained ground in several fields of research and as a model for complex brain disorders [23]. It has also been used to evaluate the pathological mechanisms of affective disorders [24]. This can be achieved through the zebrafish's response to different stimuli, which allows researchers to study biological and pharmaceutical processes [25]. From a pharmacological point of view, considering the beneficial effects, the studies presented the actions of VPA, such as promoting innovative neuroprotective, even antidiabetic and cardioprotective [26]. However, in addition to the effects in case of withdrawal of this drug, these also come with its potential side effects, such as hepatotoxicity, anemia, coagulopathy and teratogenic effects [27]. From a toxicological point of view, we are interested in clarifying the possible toxic effects that VPA can have as a result of unconscious exposure, especially from residues left in the water, at minimal doses, but also whether or not microplastics can act synergistically with it.

The main purpose of this study was to evaluate the individual effects of polyethylene, polypropylene, and valproic acid, at environmentally relevant concentrations [21], and their combined effects on the adult stage of *Danio rerio*, at behavioral and biochemical levels. We hypothesize that all three contaminants, either alone or in combination, could have a toxicological impact on the zebrafish.

## 2. Materials and Methods

### 2.1. Ethical Note

Animals were treated and maintained in accordance with the EU Commission Recommendation (2007), Directive 2010/63/EU of the European Parliament, and Council guidelines of 22 September 2010 on the accommodation, care and protection of animals used for experimental and other scientific purposes. The protocol we followed received approval of the Ethics Commission of the Faculty of Biology, "Alexandru Ioan Cuza" University, Iasi, with registration No. 343/09.02.2023.

### 2.2. Animal Maintenance

For the study, we used 30 adult zebrafish, approximately 9–10 months old, (*Danio rerio*) from an authorized local breeder. The zebrafish used in this experiment had a period under experimental laboratory conditions of 10 days, in 10 L aquariums, equipped with oxygen pumps and dechlorinated water changed daily, with the following characteristics: temperature 26 ± 1 °C, pH = 7.5, dissolved oxygen 7.20 mg $L^{-1}$, ammonia concentration < 0.004 ppm and conductivity 500 µS. After this period, zebrafish were randomly assigned to experimental groups (n = 5).

### 2.3. Experimental Design

Zebrafish were exposed to valproic acid (VPA), polypropylene (PP) and polyethylene (PE) microplastics, respectively, in combinations of PE+VPA (PEV) and PP+VPA (PPV). The concentrations used were as follows: 2 mg L$^{-1}$ for the polymeric materials, administered by daily dietary exposure, respectively 25 μM VPA, immersed in the environment for 30 min. The concentration was chosen based on the concentrations found in the environment, but also based on other environmental-related tests that were conducted [28,29]. The polymeric material was administered for 5 consecutive days, followed by the administration of VPA for 5 days in the specific groups. The diet was in line with the housing and carrying requirements, meaning that 8% of their food weight was administered, combined with the required amount of polymeric material. Behavioral response was analyzed 24 h after the first dose for all treatments and at the end of the experiment, using the EthoVisionXT 14 video tracking software (Noldus Information Technology, Wageningen, The Netherlands). To observe the joint effects of VPA-polymeric material, the following behavioral tests were used: Novel Tank Test, Social Preference Test and Aggressivity Test. At the end of the experiment, fish were sacrificed according to ethical procedures for the analysis of oxidative stress biomarkers, namely SOD, MDA, GPx and total soluble protein.

#### 2.3.1. Novel Tank Test

The experimental setup was performed following the literature suggestions [30]. It consisted of a rectangular tank, filled with 6 L of water and divided equally into upper and bottom halves. Each fish was placed individually into the tank after the designated treatment periods. To evaluate fish behavior several parameters were assessed using the EthoVision XT 16 software: distance moved, velocity, latency to reach the upper half of the tank, the time spent in the upper half, number of entries into the upper half, inactivity time, and circling behavior. In addition, for a better idea of the level of anxiety, the aquarium, seen from above, was divided into two other areas (peripheral and central). The time spent by the animal in the peripheral zone was used to calculate the anxiety index, according to Freitas et al., 2023 [31]. This index represents the time spent in the peripheral area divided by the total time tested and multiplied by 100. Increased thigmotaxis behavior, usually considered an indicator of anxiety, is associated with the presence of fish in the peripheral zone of the aquarium.

#### 2.3.2. Social Preference Test

The social preference test was performed with a T-maze consisting of two arms (left arm, right arm) and a first arm where the start box is located, based on the literature suggestions [32,33]. The maze is used to study how fish make different choices and behave in different situations, giving the subject a direct choice. To determine social behavior, several conspecifics were placed in a box in the left arm of the maze. The box was created by placing a transparent wall. The tested fish was placed in the start box at the end of the first arm. The frequency and time spent in the left arm of the maze will provide information about the social behavior of the fish, in addition to swimming performance parameters (distance moved, velocity, inactivity time) monitored with EthoVision XT 16 software.

#### 2.3.3. Aggressivity Test

The aggressivity test was performed, based on the literature suggestions [34,35], with a T-maze by closing one arm with a mirror, so that the maze consisted of three arms: one left where the mirror was placed, one right, and a main arm where the start box was located. Each fish spent 4 min in the maze where several parameters were analyzed with EthoVision XT 16 software, such as frequency and presence of the fish in the left arm where the mirror is located, swimming bursts, counterclockwise rotations, distance moved and velocity.

#### 2.3.4. Oxidative Stress Analysis

Following behavioral assessment, all animals were euthanized according to standard laboratory euthanasia procedures by immersion in cold water (2–4 °C) for at least 10 min,

until the cessation of opercular movements. The whole fish body was gently homogenized on ice and a tissue extraction buffer was added. The obtained homogenates were then centrifuged at 3500 rpm for 15 min, according to the procedure previously described by our research group [34]. The supernatant was used to determine the enzymatic activities of superoxide dismutase (SOD) and glutathione peroxidase (GPX), and quantify malondialdehyde (MDA) levels. Total soluble protein levels were assessed using Bradford method (Protein Quantification Kit—Rapid (Sigma, Taufkirchen, Germany)) and were used to calculate the specific activity of antioxidant enzymes (UE/mg TSP) and the relative quantity of MDA (umol/mg TSP). All assays were performed according to the manufacturer's recommended procedures (SOD Assay Kit (Sigma, Taufkirchen, Germany), GPx Cellular Activity Assay Kit CGP-1 (Sigma, Taufkirchen, Germany)). Malondialdehyde (MDA) levels were assessed using the thiobarbituric acid reactive substances (TBARS) method, according to a previously established protocol [36].

### 2.4. Statistical Analysis

Normality and distribution of data were determined by the Shapiro–Wilk test, using Graph Pad Prism software 9 (San Diego, CA, USA). Multiple comparisons between groups were then performed using one-way ANOVA, followed by Tukey's test. Data are expressed as mean $\pm$ SEM (standard error of the mean), and a $p < 0.05$ was considered statistically significant.

## 3. Results

### 3.1. Behavior Analysis

#### 3.1.1. Novel Tank Test

In PE treatments, distance moved during the tests increased with increasing doses of PE and subsequent doses of VPA, although no statistically significant differences were found (Figure 1A). The same trend was observed for velocity. However, in this case, the difference between control and PE VPA D5 was statistically significant ($p < 0.05$, Figure 1B). No statistically significant differences were found for the time of inactivity (Figure 1C).

The number of entries in the upper part of the aquarium was surprisingly different. There was a statistically significant value ($p < 0.05$) between control and PE VPA 24H and $p < 0.01$ between control and PE VPA D5. Statistically significant differences ($p < 005$) were also found between PE D5 and PE VPA 24H, respectively (Figure 1D). Latency to reach the upper half was also highly variable, and statistical differences were found between the control group and PE 24H and PE VPA 24H, respectively ($p < 0.05$). Other significant differences were found between PE 24H and PE VPA D5, and PE VPA 24H and PE VPA D5 ($p < 0.01$, Figure 1E). Time spent in the upper part of the aquarium showed the same increasing trend as distance and velocity. A highly significant difference was recorded between PE VPA D5 and the other groups ($p < 0.001$, Figure 1F). Regarding thigmotaxis, a high significant difference was recorded between the control group and the PE VPA D5 group ($p < 0.01$, Figure 1G). The same trend was observed for the anxiety index. There were significant differences between control and PE VPA D5 ($p < 0.05$, Figure 1H).

In the PP group, distance moved and velocity decreased over the duration of the tests with administered doses of PP, with subsequent VPA treatments significantly increasing them compared to PP 24H ($p < 0.05$, Figure 2A,B). Inactivity time in PP VPA 24H was significantly increased compared to control ($p < 0.001$) and compared to PP 24H ($p < 0.05$) (Figure 2C). The number of entries into the upper part of the aquarium was significantly decreased compared to control ($p < 0.001$) in PP 24H, PP D5 ($p < 0.01$) and PP VPA 24H ($p < 0.05$). There was a significant increase between PP 24H and PP VPA D5 ($p < 0.05$) (Figure 2D). For the number of entries into the upper half, there were significant differences between treatments and doses as follows: PP 24H vs. PP VPA 24H ($p < 0.01$), PP 24H vs. PP VPA D5 ($p < 0.05$), PP D5 vs. PP VPA 24H ($p < 0.001$) and PP D5 vs. PP VPA D5 ($p < 0.01$) (Figure 2E). The time spent in the upper half was significantly increased in the PP 24H and PP D5 groups, respectively, compared with the control group ($p < 0.05$, Figure 2F).

Thigmotaxis and the anxiety index were variable. Both increased significantly between PP 24H and PP VPA D5 (Figure 2G,H).

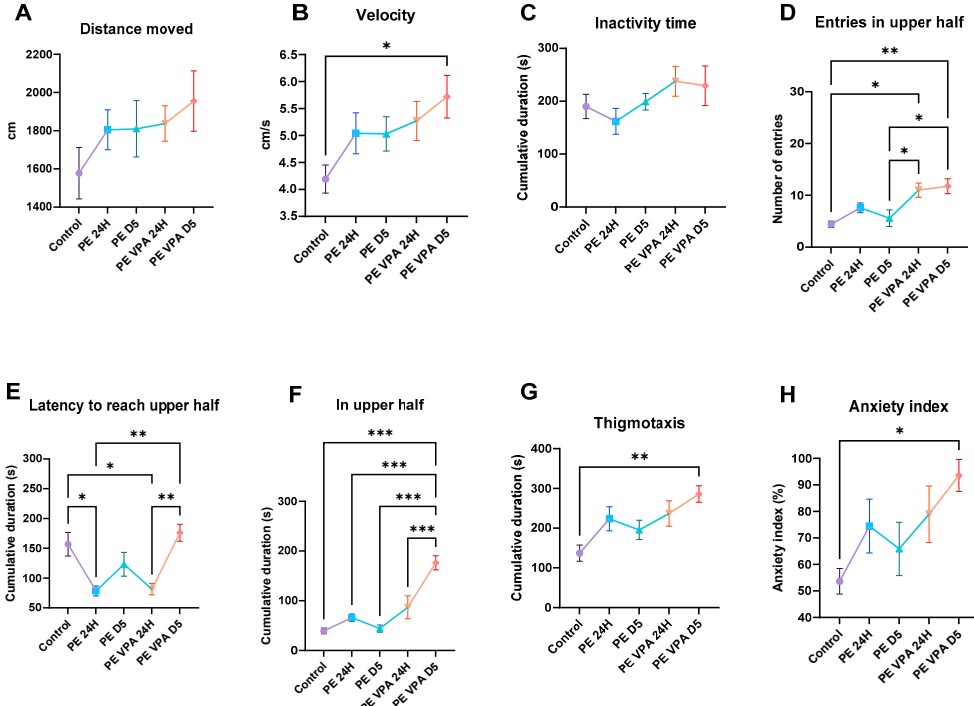

**Figure 1.** Graphical representation of behavior patterns in PE and PEV treatments in the Novel Tank Test. Data are expressed as mean ± SEM, and a significance level of $p < 0.05$ was considered statistically significant (*), $p < 0.01$ was considered very significant (**), and $p < 0.0001$ was considered higher significant (***).

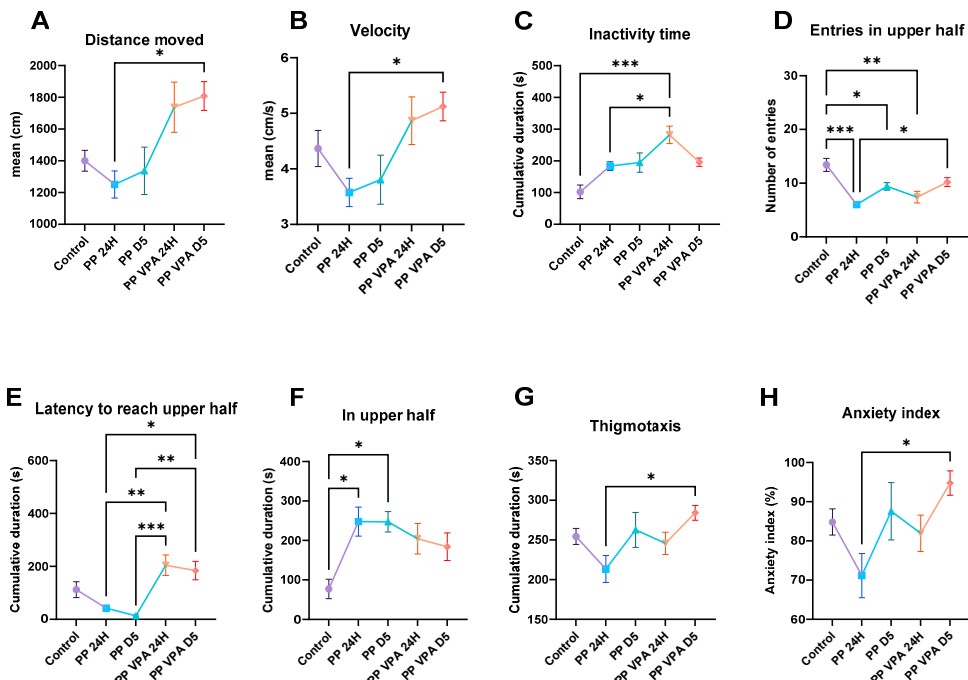

**Figure 2.** Graphical representation of behavior patterns in PP and PPV treatments in the Novel Tank Test. Data are expressed as mean ± SEM, and a significance level of $p < 0.05$ was considered statistically significant (*), $p < 0.01$ was considered very significant (**), and $p < 0.0001$ was considered higher significant (***).

### 3.1.2. Social Preference Test

Regarding the social preference test in PE treatments, the distance moved decreased, with significant differences between control and PE D5, respectively, PE VPA 24H ($p < 0.05$) and PE VPA D5 ($p < 0.01$) (Figure 3A). Velocity significantly decreased in PE VPA D5 compared to control ($p < 0.05$, Figure 3B). Inactivity time significantly increased in PE VPA D5 compared to control ($p < 0.05$, Figure 3C). The frequency of left arm entry was decreased in PE D5 ($p < 0.05$) and PE VPA D5, respectively, compared with control ($p < 0.01$, Figure 3D). The presence of fish in the left arm with conspecifics was recorded in all treatment cases, with no statistical differences for this parameter (Figure 3E).

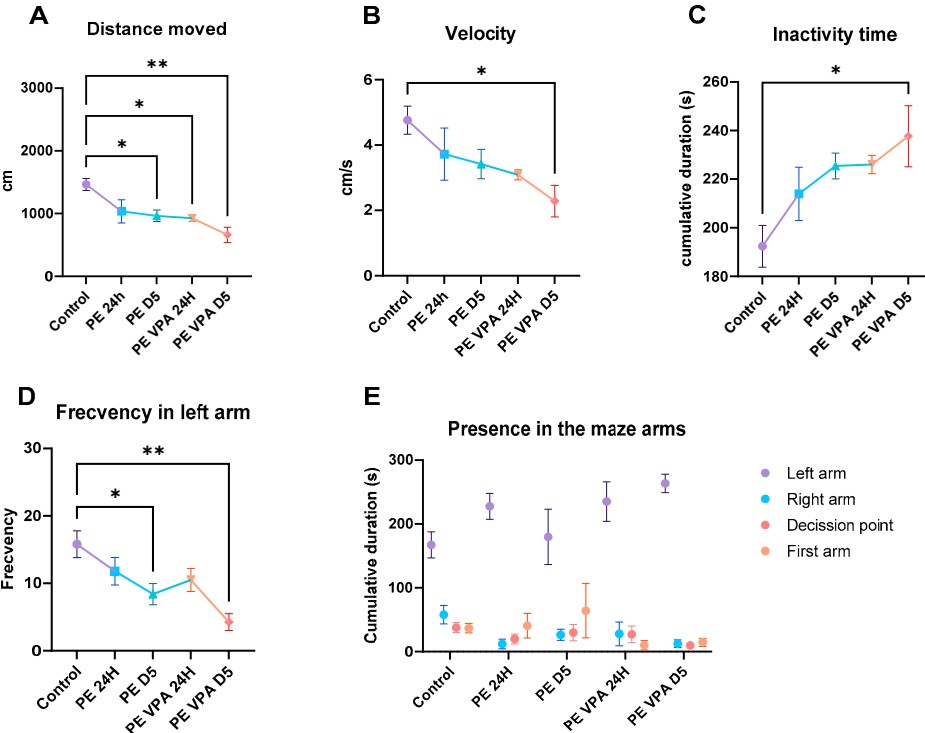

**Figure 3.** Graphical representation of behavior patterns in PE and PEV treatments in the Social Preference Test. Data are expressed as mean ± SEM, and a significance level of $p < 0.05$ was considered statistically significant (*), $p < 0.01$ was considered very significant (**).

PP treatments showed no significant differences for four out of five parameters monitored: distance moved (Figure 4A), velocity (Figure 4B), inactivity time (Figure 4C) and frequency in the left arm (Figure 4D). Significant differences were recorded for presence in the left, namely control versus PP D5 and PP D5 versus PP VPA 24H ($p < 0.05$, Figure 4E).

### 3.1.3. Aggressivity Test

In the PE treatments, distance moved, and velocity maintained the same decreasing trend of the values recorded during the treatments. There were statistically significant differences between control and PP VPA D5 ($p < 0.05$, Figure 5A,B). Presence in the left arm increased significantly in PE D5 compared to control ($p < 0.01$) and decreased significantly in PE VPA 24H compared to PE D5 ($p < 0.05$, Figure 5C). Frequency in the left arm significantly increased in PE D5 compared to control ($p < 0.05$) and significantly decreased in PE VPA D5 compared to PE D5 ($p < 0.05$, Figure 5D). The swim burst significantly decreased in PE VPA D5 compared to the control and in PE 24H ($p < 0.05$, Figure 5E). Counterclockwise rotations significantly increased in PE D5 vs. control ($p < 0.01$) and vs. PE 24H ($p < 0.001$). Significant decreases after VPA treatments were observed between PE D5 vs. PE VPA 24H and PE VPA D5, respectively ($p < 0.001$, Figure 5F).

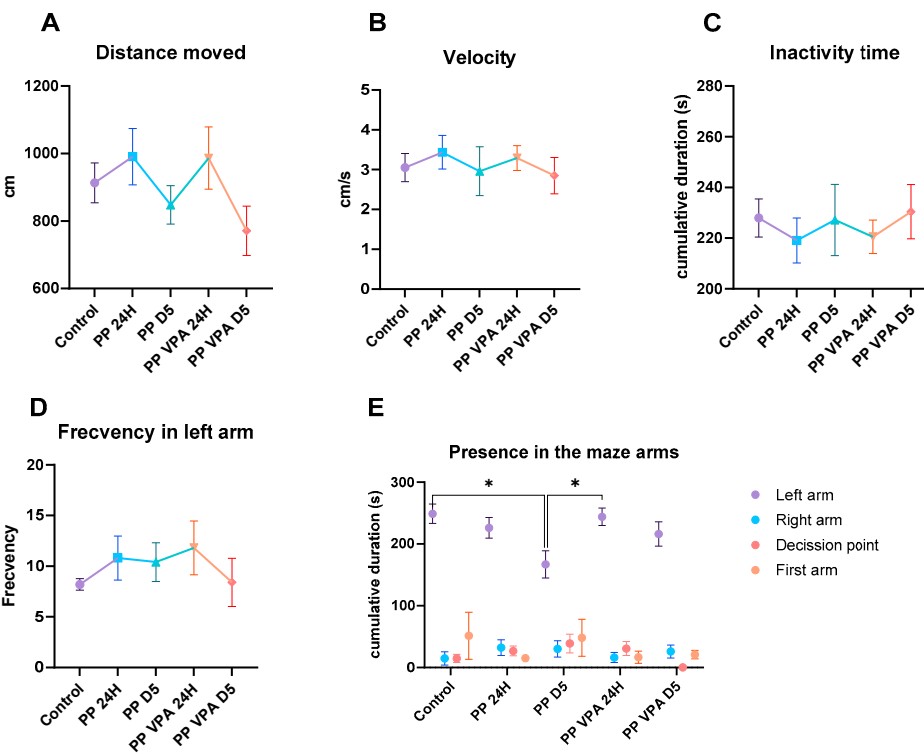

**Figure 4.** Graphical representation of behavior patterns in PP and PPV treatments in the Social Preference Test. Data are expressed as mean ± SEM, and a significance level of $p < 0.05$ was considered statistically significant (*).

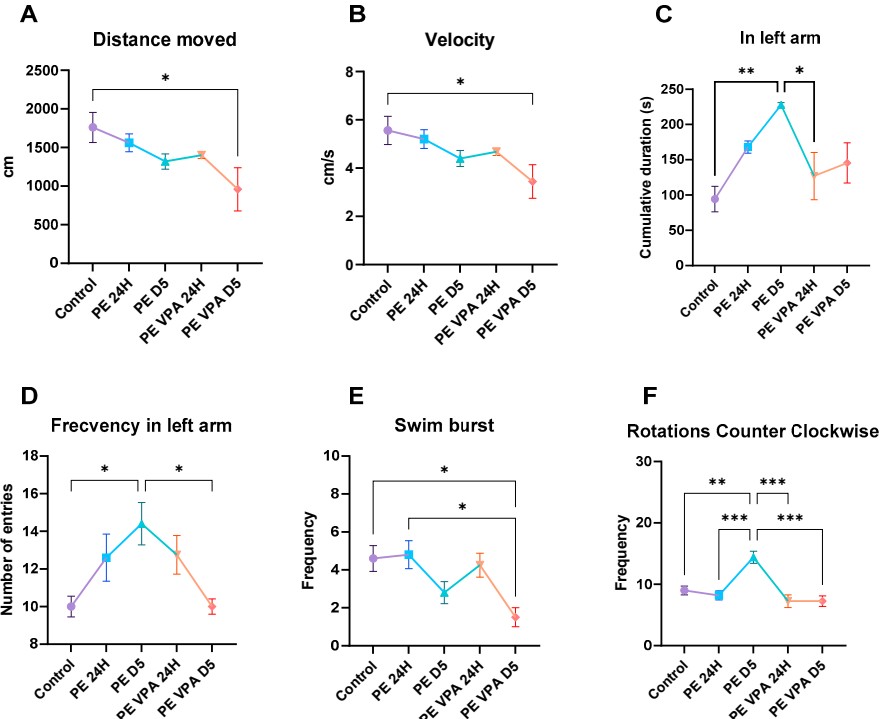

**Figure 5.** Graphical representation of behavior patterns in PE and PEV treatments in Aggressivity Test. Data are expressed as mean ± SEM, and a significance level of $p < 0.05$ was considered statistically significant (*), $p < 0.01$ was considered very significant (**), and $p < 0.0001$ was considered higher significant (***).

PP treatments showed no significant differences for four out of five parameters monitored: distance moved (Figure 6A), velocity (Figure 6B), presence in the left arm (Figure 6C), and swim burst (Figure 4E). Significant differences were observed for left arm frequency, namely between control and PP VPA D5 ($p < 0.05$, Figure 6D). For the counterclockwise rotations, significant increases were registered as follows: control vs. PP 24H ($p < 0.05$), control vs. PP VPA 24H ($p < 0.01$), and PP D5 vs. PP VPA 24H ($p < 0.05$, Figure 6F).

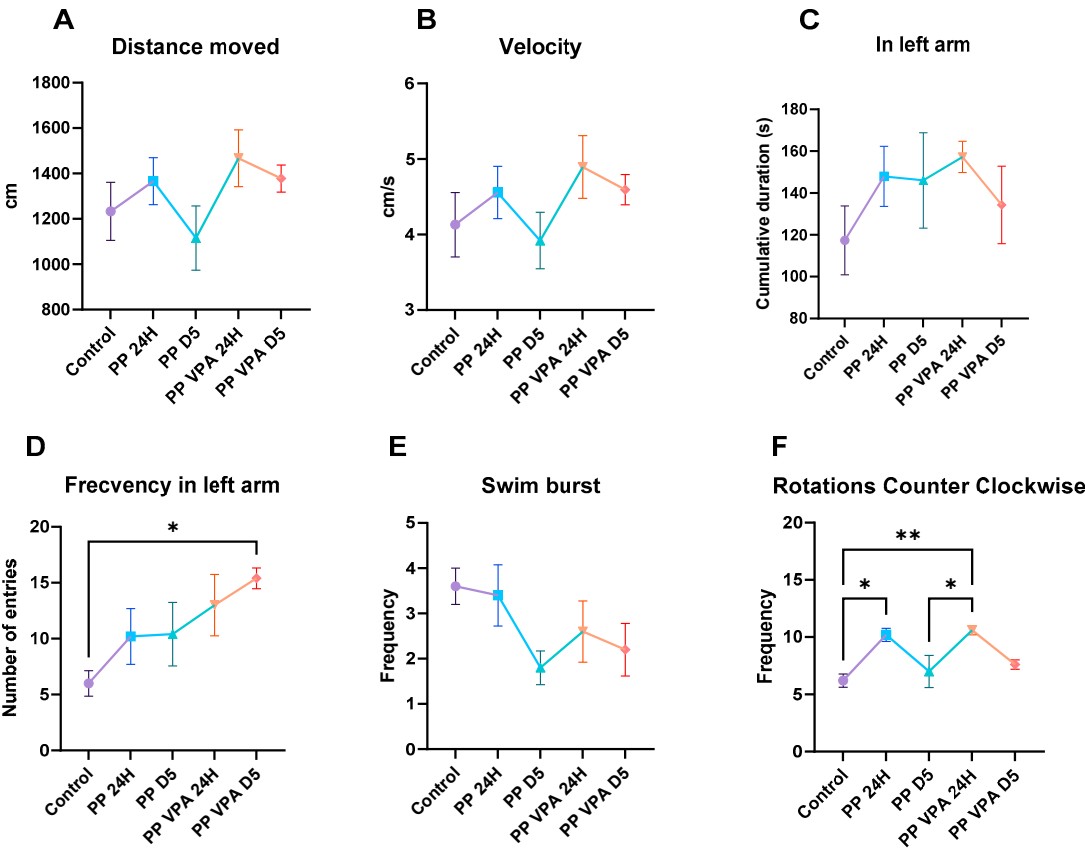

**Figure 6.** Graphical representation of behavior patterns in PP and PPV treatments in Aggressivity Test. Data are expressed as mean ± SEM, and a significance level of $p < 0.05$ was considered statistically significant (*), $p < 0.01$ was considered very significant (**).

### 3.2. Oxidative Stress Analysis

We found a suggestive increased specific SOD activity following exposure to PE and PPV compared to the control group. Significant difference was found in the following situations: PP vs. PE ($p < 0.05$), PP vs. PPV ($p < 0.05$), PE vs. PEV ($p < 0.05$), VPA vs. PPV ($p < 0.05$), PPV vs. PEV ($p < 0.05$) (Figure 7A). Regarding the antioxidant activity of GPx, we observed that it was significantly lower after exposure to PE compared to the control group ($p < 0.01$). Another significant difference was found between PP vs. PE ($p < 0.01$) and PE vs. VPA ($p < 0.05$, Figure 7B). Commonly, we observed that MDA levels increased following treatments, compared to controls, except for VPA exposure which resulted in the opposite effect. Statistically significant changes in MDA content were observed for CTR vs. PPV ($p < 0.01$), PP vs. VPA ($p < 0.01$), PP vs. PPV ($p < 0.05$), VPA vs. PEV ($p < 0.05$), and VPA vs. PPV ($p < 0.001$) comparisons (Figure 7C).

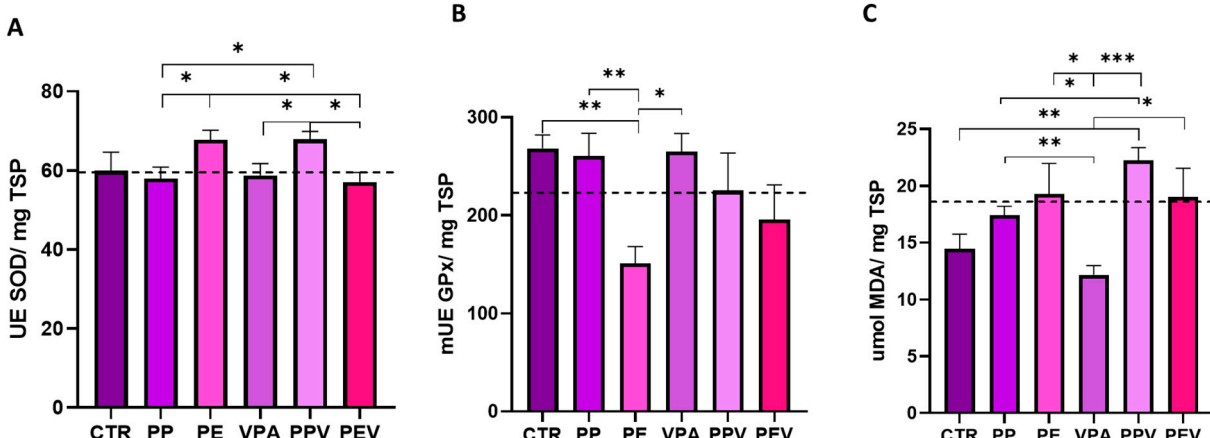

**Figure 7.** Graphical representation of the levels of oxidative stress markers: SOD-specific activity (UE SOD/mg TSP) (**A**), GPx specific activity (mUE GPx/mg TSP) (**B**), and MDA relative content (umol MDA/mg TSP) (**C**). Data are expressed as mean ± SEM (n = 5/per group, * $p < 0.05$, ** $p < 0.01$, *** $p < 0.001$ in two-tailed *t*-test). TSP = total soluble proteins.

## 4. Discussion

*Danio rerio* has been established as a suitable animal model for pharmaceutical studies [37], especially due to the possibility of immersion in water of the investigated substances and the high potential of zebrafish to absorb these compounds [38]. Zebrafish has proven to be an excellent tool due to its high genetic similarity to humans [39], thus motivating the study of some human diseases using the *Danio rerio* animal model. In this sense, the zebrafish is an appropriate model for biomedical research [40]. Due to the behavioral characteristics, and homology observed with humans with autism spectrum disorder (ASD), the zebrafish is also a relevant model for the study this disease [37]. Studies have shown that VPA can induce ASD-like symptom behavior in zebrafish juveniles at a concentration of 48 μM VPA [41]. This applies to both larval and adult zebrafish [42]. Other studies have shown that valproic acid can negatively affect embryonic vasculature from exposure concentrations starting at 2.5 μM [43], but can also affect larval social behavior [44]. Exposure between 0.33 and 4.5 days post-fertilization (dpf) to 10 μM VPA was identified as an effective concentration to induce an early and persistent ASD-like phenotype in zebrafish, as embryonic exposure to VPA also reduces survival, induces malformations and delays hatching in a dose- and time-dependent manner, also triggering hyperactivity, anxiety-like behavior and social deficits [42].

Regarding plastic pollution, limiting the flow of plastic from rivers to marine ecosystems is an important part of reducing the amount of plastic in the environment [45]. The spread of polymers in both rivers and oceans can be stopped by effective management that removes plastics directly from their source. Plastic pollution has become a major environmental concern due to its extensive use and fast spread [46]. Microplastics in freshwater ecosystems and marine environments can cause both physical and histological damage [47]. What is more more, PE and PP are the most common plastics found in the marine environment [48].

In terms of behavior analysis, in the case of the Novel Tank test, considering the distance moved and velocity in the case of the PE-VPA treatments, PE has an anxiolytic effect immediately after the first dose, and it increases with the dose. The treatment with VPA increased an anxiogenic effect from the first dose and by day five, especially by the increased inactivity time. The latency to reach the upper part of the aquarium is lower after the first dose of PE treatment and then increases with the number of doses, with fish preferring the upper part of the aquarium, especially after VPA administration. In the case of PP-VPA treatments, PP induces the opposite behavior to PE. Thus, from the first dose

of PP, the fish prefer the upper part of the aquarium. This trend is maintained until the 5th dose of PP. This also holds true when treated with VPA.

In the case of PE, we noticed that starting from the first dose of VPA, the latency decreased and the fish explored the entire aquarium again. However, at the first dose of PE and at the 5th dose of VPA thigmotaxis was most pronounced. In the case of PP, the thigmotaxis is more evident only at the 5th dose and is maintained during VPA treatment. VPA treatment significantly increases the total exploratory capacity expressed by the distance moved of the fish.

Anxiety-like behavior was triggered in different studies, for example in rats exposed to 600 mg kg$^{-1}$ VPA [49] or 70 dpf zebrafish exposed to 48 μM VPA [40]. The most obvious results were observed when analyzing the sociability test. The first dose of PE increased the presence of fish close to conspecifics. This presence began to decrease over the course of treatment. In the case of the first VPA treatment, the presence of fish near conspecifics is much higher and maintains an upward trend until the 5th day of treatment. During both treatments, distance moved and velocity were steadily decreasing. Regarding social preference, as in the case of PE, the plastic treatment decreases sociability until the 5th day. However, VPA has a great influence, so that from the first day of treatment with VPA, the presence of fish is majority in the arm where the conspecifics are located. Sociability is more affected at the fifth dose of PP. In this case, the affected mechanisms are more related to swimming performance.

In our study, VPA does not induce social impairments or hyperactivity deficits [42] or other ASD-like events. VPA increases presence in the left arm but also increases inactivity time, indicating a possible anxiety mechanism. When testing aggressiveness, velocity and distance moved show similar trends. In the case of the first dose of PE, they continue to decrease until the fifth dose. The first dose of VPA produces a slight increase in these two parameters, but by the 5th dose both decrease to almost half of the control value. On the other hand, the first dose of PE induces more aggressive behavior compared to the control, with an increasing trend up to the 5th dose. In the case of PP, total distance moved and velocity decreased first and started to increase during VPA treatments. The presence of fish near the mirror is higher from the first day of treatment with PP and is maintained until the 5th day of treatment with VPA. The first dose of VPA increased this presence. However, it decreased in direct proportion to the increase in dose. For the PE group, the first dose of VPA significantly decreases the presence of fish near the mirror, but by day 5, this presence increases again. Also, counterclockwise rotations, which are a marker of aggressiveness [34], were observed with a high frequency compared to the control group in the first 24 h after PP treatment and at 24 h after VPA treatment, reaching limits significantly lower until the 5th day of treatment. Significant results were obtained in the PE-VPA treatments. In the PE group, the highest increase was observed on day 5, while the other groups maintained the same frequency as the control group. This reinforces the idea that PE has an anxiogenic effect on fish behavior. The frequency of swim bursts decreases with the PP treatments and increases with the VPA treatment. In this case, PP produces a non-aggressive behavior, but the situation changes radically from the first dose of VPA. Moreover, the frequency of swim bursts was lower at the 5th dose of PE compared to the first, but treatment with VPA decreases this frequency by the 5th dose.

It is worth noting that in the case of VPA treatments, behavioral differences appear with increasing doses [40,42,50]. In the case of our study, the administration of VPA after intoxication with polymeric materials indicates the same thing, especially since depending on the polymeric material in some cases the first dose is the one that alleviates the effects and then they worsen, or only the fifth dose is the one that helps. In this regard, if VPA can be used for the treatment of polymer intoxication, it is important to know what the purpose of VPA administration would be. However, in the case of exposure to these pollutants under the conditions simulated in this study, the effects are visible and have a negative impact.

Superoxide dismutase (SOD) is an antioxidant enzyme that plays a critical role in the neutralizing superoxide radicals, being the first antioxidant enzyme within the cellular enzymatic defense against oxidative stress. It is currently thought that increased levels of SOD are the results of an adaptive mechanism that counteracts the pro-oxidative status and thus protects cells from potential damage. This response is often seen in a variety of physiological or pathological conditions where the increased production of reactive oxygen species occurs [51]. In our study, we observed increased levels of SOD following PE and PP+VPA exposure. Other studies have shown significantly increased levels of SOD and decreased levels of GPx in the liver of adult zebrafish exposed to PP for 21 consecutive days [43]. However, no significant changes were observed in the brain of exposed fish [52]. Lower levels of GPx may indicate a reduced ability to neutralize free radicals, which may make cells more susceptible to oxidative damage. This may be associated with increased oxidative stress, which has been implicated in several health problems, including chronic inflammation, neurodegenerative disease, and cardiovascular disease [53]. In our study, lower levels of GPx were found in the PE group. Similarly, a 96 h exposure to PE beads of different sizes reduced GPx activity in the brain and liver of adult zebrafish [54]. Elevated levels of malondialdehyde (MDA) are often considered a marker of lipid peroxidation. MDA indicates oxidative damage to cell membranes. Elevated MDA levels may be associated with increased oxidative stress. MDA has been implicated in several health conditions, including inflammation, cardiovascular disease, and neurodegenerative disorders [55]. In our study, all groups except VPA showed a high level of MDA compared to the control group. PP significantly elevated malondialdehyde (MDA) levels in the stomach in a 28-day study on male zebrafish. Moreover, combination of triclosan and PP significantly aggravated oxidative stress and lipid peroxidation in the liver as well as has enhanced neurotoxicity in the brain [56]. In another 21-day study, PP-MPs resulted in increased levels of MDA in gills and liver cells compared to controls, indicating a dose-dependent effect [57]. In our study, only VPA was shown to lower MDA levels, similar to another study that also found lower MDA levels when adult zebrafish were exposed to 0.5 mg mL$^{-1}$ VPA. Furthermore, compared to either substance alone, the combination of rotenone and VPA showed an increased level of MDA [34]. The same was found when we combined PP and VPA.

One aspect that needs to be taken into account is the shape of the microplastics used for the study and their capacity for transport. A study published in 2023 suggests that fibers are more effective at transporting over long distances than other shapes [58]. In our study, we have used both fibers as well as irregular flat shapes, but specific studies on this aspect should be taken into consideration in the future, along with another different analyses such as hormones or histology aspects to fully understand the mechanism of action of all the compounds of these polymeric materials and pharmaceuticals.

## 5. Conclusions

In conclusion, the two types of plastic have very different effects: PE induces anxious behavior, while PP induces hyperactive behavior. Regarding aggressive behavior, PE treatment resulted in an increased manifestation of this behavioral trait compared to PP treatment. Conversely, an increased level of aggressive behavior was observed when VPA was administered. Sociability is the parameter most affected by the presence of polymers, both of which produce antisocial behavior. Both treatment groups show an evident social preference. However, a possible anxiety mechanism is actually hidden, especially due to increased inactivity time and influence on swimming performance from the first dose of VPA administered. All three pollutants, alone or in cocktail, had a marked influence on GPx and MDA in terms of oxidative stress. In light of all the results of our study, our hypothesis has been confirmed.

However, future research is needed on both the negative effects of the toxic cocktail and the potential of VPA to mitigate the negative effects of microplastics toxicity.

**Author Contributions:** Conceptualization, methodology, software, writing—original draft preparation A.S. and I.-A.C., investigation A.S., A.-S.C., I.-M.B. and I.-A.C., writing—review and editing, visualization, supervision A.S.C. and M.N.N. All authors have read and agreed to the published version of the manuscript.

**Funding:** This research received no external funding.

**Institutional Review Board Statement:** The animal study protocol was approved by the Institutional Ethics Committee of the Faculty of Biology, "Alexandru Ioan Cuza" University, Iasi with No. 343/09.02.2023.

**Informed Consent Statement:** Not applicable.

**Data Availability Statement:** Data are contained within the article.

**Conflicts of Interest:** The authors declare no conflicts of interest.

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
