# Peer review of "Toxicological Response of Zebrafish Exposed to Cocktails of Polymeric Materials and Valproic Acid"

_sustainability, doi:10.3390/su16052057_

Round 1

Reviewer 1 Report

Comments and Suggestions for Authors

First of all, I would like to thank you very much for choosing our journal for your article. It is a very successful and meticulously prepared article. If you answer the questions I have asked, I would like to read the article again.

- Can you provide more detailed information about the ethical review process for your study? Specifically, how does the Ethics Commission of the Faculty of Biology at “Alexandru Ioan Cuza” University assess studies involving animals, and what specific criteria were used to approve this study?

- Were any forms of environmental enrichment provided in the aquariums to promote natural behaviors and well-being of the zebrafish during the maintenance and experimental periods?

- How were the dosages of valproic acid (VPA), polypropylene (PP), and polyethylene (PE) determined for this study? Please provide a rationale or references supporting these specific concentrations.

- Have the Novel Tank Test, Social Preference Test, and Aggressivity Test been validated for use in assessing the specific behaviors you are investigating in zebrafish? Please provide references or data supporting their validity.

- Could you provide more details on the procedures for homogenizing fish tissues and preparing samples for oxidative stress analysis? Specifically, how was variability in sample processing minimized?

- You mentioned using Shapiro-Wilk test for normality and one-way ANOVA for multiple comparisons. Were there any instances where the data did not meet the assumptions of these tests, and if so, how were these instances handled?

- Were there any potential confounding factors identified that could have influenced the results, and how were these addressed in the study design or analysis?

- Given the detailed description of animal treatment, could you elaborate on the specific ethical considerations taken into account for the euthanasia process, especially regarding the choice of cold water immersion? How does this method align with the latest recommendations for humane endpoints in zebrafish research?

- In the social preference and aggressivity tests, how were the control conditions defined, particularly in terms of environmental and social stimuli? Were any additional controls used to account for potential stressors unrelated to the experimental treatments?

- For the statistical analyses conducted, particularly the use of Shapiro-Wilk test and one-way ANOVA, were there any instances where the data did not meet the assumptions required for these tests? If so, how were these challenges addressed?

- Regarding the observed behavioral changes in the Novel Tank, Social Preference, and Aggressivity Tests, how do the authors interpret the significance of these changes in the context of potential environmental or pharmacological impacts? Specifically, how might these behaviors correlate with known effects of pollutants or pharmacological agents on wildlife or human health?

- Could the authors elaborate on the limitations of their study, particularly regarding the extrapolation of results to natural environments or human health? What future research directions do the authors recommend based on their findings?

- Given the increasing concern about microplastic pollution and chemical contaminants in aquatic environments, how do the authors envision their findings contributing to environmental protection efforts or influencing regulatory policies?

Author Response

First, I would like to thank you very much for choosing our journal for your article. It is a very successful and meticulously prepared article. If you answer the questions I have asked, I would like to read the article again.

~Dear Reviewer 1, thank you for your appreciation. Below you will find the answers to each question, we hope that the context is now much clearer and better highlighted. ~

- Can you provide more detailed information about the ethical review process for your study? Specifically, how does the Ethics Commission of the Faculty of Biology at “Alexandru Ioan Cuza” University assess studies involving animals, and what specific criteria were used to approve this study?

~The ethics committee evaluates all work protocols in accordance with the EU Commission Recommendation (2007), Directive 2010/63/EU of the European Parliament, and Council guidelines of 22 September 2010 on the accommodation, care and protection of animals used for experimental and other scientific purposes. The main criteria used for assessing that the ethical considerations are met is the 3 Rs principles (replace, reduce, and refine). In the sense that if a study does not require the mandatory use of animals, the methods will be modified; if the study does require the use of animal models, then the minimum number of specimens should be used to obtain significant results; and make sure that the animal suffer as little as possible for each experimental approach. ~

- Were any forms of environmental enrichment provided in the aquariums to promote natural behaviors and well-being of the zebrafish during the maintenance and experimental periods?

~According to good practice guidelines, the aquariums were not equipped with any environmental enrichments (e.g. substrate, special aquarium trees, shells). Instead, we made sure that they had enough space for swimming, water` parameters were kept within proper limits, oxygen levels were maintained throughout the whole experiment, and they were fed accordingly to the guidelines (e.g. Aleström P, D'Angelo L, Midtlyng PJ, et al. Zebrafish: Housing and husbandry recommendations. Lab Anim. 2020;54(3):213-224. doi:10.1177/0023677219869037; Westerfield, M. (2000). The zebrafish book. A guide for the laboratory use of zebrafish (Danio rerio). 4th ed., Univ. of Oregon Press, Eugene.).

- How were the dosages of valproic acid (VPA), polypropylene (PP), and polyethylene (PE) determined for this study? Please provide a rationale or references supporting these specific concentrations.

~ In the case of VPA, the concentration was chosen taking into account the concentrations found in the environment (Al-Khazrajy, O.S.A.; Boxall, A.B.A. Risk-Based Prioritization of Pharmaceuticals in the Natural Environment in Iraq. Environmental Science and Pollution Research 2016, 23, 15712–15726, doi:10.1007/s11356-016-6679-0; Perazzolo, C.; Morasch, B.; Kohn, T.; Magnet, A.; Thonney, D.; Chèvre, N. Occurrence and Fate of Micropollutants in the Vidy Bay of Lake Geneva, Switzerland. Part I: Priority List for Environmental Risk Assessment of Pharmaceuticals. Environ Toxicol Chem 2010, 29, 1649–1657, doi:10.1002/etc.221; Yu, J.T.; Bisceglia, K.J.; Bouwer, E.J.; Roberts, A.L.; Coelhan, M. Determination of Pharmaceuticals and Antiseptics in Water by Solid-Phase Extraction and Gas Chromatography/Mass Spectrometry: Analysis via Pentafluorobenzylation and Stable Isotope Dilution. Anal Bioanal Chem 2012, 403, 583–591, doi:10.1007/s00216-012-5846-5; Yu, J.T.; Bouwer, E.J.; Coelhan, M. Occurrence and Biodegradability Studies of Selected Pharmaceuticals and Personal Care Products in Sewage Effluent. Agric Water Manag 2006, 86, 72–80, doi:10.1016/j.agwat.2006.06.015.) Moreover, another reason for choosing this concentration is other environmentally relevant tests that have been done on this compound in the literature, as opposed to animal models. Regarding the polymeric materials, we chose a concentration usually used for zebrafish animal models in the literature. ~

- Have the Novel Tank Test, Social Preference Test, and Aggressivity Test been validated for use in assessing the specific behaviors you are investigating in zebrafish? Please provide references or data supporting their validity.

~These tests are considered to be standard and validated within the discovery and implementation of this animal model in more and more studies. Some references for using the tests are listed below:

Novel Tank Test

Cachat, J. M., Canavello, P. R., Elkhayat, S. I., Bartels, B. K., Hart, P. C., Elegante, M. F., Beeson, E. C., Laffoon, A. L., Haymore, W. A. M., Tien, D. H., Tien, A. K., Mohnot, S., & Kalueff, A. v. (2011). Video-aided analysis of zebrafish locomotion and anxiety-related behavioral responses. Neuromethods, 51, 1–14. https://doi.org/10.1007/978-1-60761-953-6_1

Aggressivity Test

Way, G. P., Southwell, M., & McRobert, S. P. (2016). Boldness, Aggression, and Shoaling Assays for Zebrafish Behavioral Syndromes. Journal of Visualized Experiments : JoVE, 2016(114), 54049. https://doi.org/10.3791/54049

Social Preference Test

Ogi, A., Licitra, R., Naef, V., Marchese, M., Fronte, B., Gazzano, A., & Santorelli, F. M. (2021). Social Preference Tests in Zebrafish: A Systematic Review. Frontiers in Veterinary Science, 7, 590057. https://doi.org/10.3389/FVETS.2020.590057/FULL

Saverino, C., & Gerlai, R. (2008). The social zebrafish: Behavioral responses to conspecific, heterospecific, and computer animated fish. Behavioural Brain Research, 191(1), 77–87. https://doi.org/10.1016/J.BBR.2008.03.013

Moreover, these tests have also been used within our research group since we started working on this animal model, some examples are listed below:

Ilie, O.-D., Duta, R., Jijie, R., Nita, I.-B., Nicoara, M., Faggio, C., Dobrin, R., Mavroudis, I., Ciobica, A., & Doroftei, B. (2022). Assessing Anti-Social and Aggressive Behavior in a Zebrafish (Danio rerio) Model of Parkinson’s Disease Chronically Exposed to Rotenone. Brain Sciences, 12(7), 898. https://doi.org/10.3390/brainsci12070898

Robea, M.A.; Jijie, R.; Nicoara, M.; Plavan, G.; Ciobica, A.S.; Solcan, C.; Audira, G.; Hsiao, C.-D.; Strungaru, S.-A. Vitamin C Attenuates Oxidative Stress and Behavioral Abnormalities Triggered by Fipronil and Pyriproxyfen Insecticide Chronic Exposure on Zebrafish Juvenile. Antioxidants 2020, 9, 944. https://doi.org/10.3390/antiox9100944

            We updated also, these references in the text, to create a clearer context. ~

- Could you provide more details on the procedures for homogenizing fish tissues and preparing samples for oxidative stress analysis? Specifically, how was variability in sample processing

~For the homogenization of the samples, a triphosphate extraction buffer was used in a mass:volume ratio of 1:10, then to make sure that this ratio did not interfere with the results of the determinations, a Chi Test was applied, which validated the used ratio

- You mentioned using Shapiro-Wilk test for normality and one-way ANOVA for multiple comparisons. Were there any instances where the data did not meet the assumptions of these tests, and if so, how were these instances handled?

~Shapiro-Wilk test for normality, is a standard test run before doing ANOVA in the GraphPrism program, used for validation, if the data does not show normality, then the program announces. In our case, the data passed the normality test, not interfering in any way with the data. ~

- Were there any potential confounding factors identified that could have influenced the results, and how were these addressed in the study design or analysis?

~Potential factors of confounding for the results, would be the synergistic action and what exactly influenced it, this factor was excluded by creating control groups for each individual compound. Besides the possible synergistic action of the compounds, we have not identified any other relevant possible confounding factors as the experiments were strictly controlled and tried to eliminate as much as possible any external influence and any non-intended internal influence. ~

- Given the detailed description of animal treatment, could you elaborate on the specific ethical considerations taken into account for the euthanasia process, especially regarding the choice of cold-water immersion? How does this method align with the latest recommendations for humane endpoints in zebrafish research?

~This euthanasia protocol is currently accepted internationally and as part of the group we considered it to be the one that produces the least suffering to the fish, since any other protocol would have involved the administration of substances that could have interfered with our experiments, in particular with the biochemical analyses (References: https://oacu.oir.nih.gov/system/files/media/file/2023-08/b17_zebrafish.pdf ; Wallace CK, Bright LA, Marx JO, Andersen RP, Mullins MC, Carty AJ. Effectiveness of Rapid Cooling as a Method of Euthanasia for Young Zebrafish (Danio rerio). J Am Assoc Lab Anim Sci. 2018;57(1):58-63.). ~

- In the social preference and aggressivity tests, how were the control conditions defined, particularly in terms of environmental and social stimuli? Were any additional controls used to account for potential stressors unrelated to the experimental treatments?

~Both of the mentioned tests are performed according to the standard procedure, already mentioned before; the control group underwent the same tests as the treatment ones. ~

- For the statistical analyses conducted, particularly the use of Shapiro-Wilk test and one-way ANOVA, were there any instances where the data did not meet the assumptions required for these tests? If so, how were these challenges addressed?

~In the case of the present study, we have not encountered any challenges in terms of meeting the assumptions of the statistical tests.

- Regarding the observed behavioral changes in the Novel Tank, Social Preference, and Aggressivity Tests, how do the authors interpret the significance of these changes in the context of potential environmental or pharmacological impacts? Specifically, how might these behaviors correlate with known effects of pollutants or pharmacological agents on wildlife or human health?

~In general, behavioral changes are associated to a neurotoxic effect (References: K.B. Tierney. Behavioural assessments of neurotoxic effects and neurodegeneration in zebrafish. Biochimica et Biophysica Acta 1812 (2011) 381–389; Dutra Costa, B.P.; Aquino Moura, L.; Gomes Pinto, S.A.; Lima-Maximino, M.; Maximino, C. Zebrafish Models in Neural and Behavioral Toxicology across the Life Stages. Fishes 2020, 5, 23. https://doi.org/10.3390/fishes5030023; Desai JK, Trangadia BJ, Patel UD, Patel HB, Kalaria VA, Kathiriya JB. Neurotoxicity of 4-nonylphenol in adult zebrafish: Evaluation of behaviour, oxidative stress parameters and histopathology of brain. Environ Pollut. 2023;334:122206. doi:10.1016/j.envpol.2023.122206), which is why studies at the biochemical level regarding oxidative stress have been made. Of course, for the complete understanding of all aspects related to these effects as well as their mechanism of action, detailed future studies are needed, this study representing a start in this field. ~

- Could the authors elaborate on the limitations of their study, particularly regarding the extrapolation of results to natural environments or human health? What future research directions do the authors recommend based on their findings?

~As in any study in the world of research, this one also has some limitations, in all studies with animals there is also the risk that the animals will not cooperate, for example, but fortunately this was not the case this time with us. A limitation, especially with human health, is the fact that this is only the first study in a series of future studies in which we hope more researchers will join us, because it is important to study each branch and aspect for a full understanding of the mechanism of action. An extraordinary limitation in our case is the fact that we cannot know for sure which chemical compound from the polymeric materials used was a trigger with such a big impact on the behavior side. Which compound reacts in a positive relationship with VPA is another limitation for which we do not have an answer. Our future research direction is to focus on eliminating as many of these limitations as possible, going deeper with the research, possibly testing compound by compound to bring to light the trigger and the difference in action of these compounds, especially in relation with the pharmaceuticals so present in the environment. Of course, it is a difficult job that can take years, but we are convinced that this is only the beginning and we hope that more researchers will join, especially after the publication of this article. ~

- Given the increasing concern about microplastic pollution and chemical contaminants in aquatic environments, how do the authors envision their findings contributing to environmental protection efforts or influencing regulatory policies?

~First of all, this research brings an alarm signal about what effects the synergy of these compounds might have. The regulatory policies regarding the protection and management of the environment could come with implementations in this regard to reduce these types of pollution, by specialized personnel in this sense. The purpose of this work is to bring to light the possible effects but also to raise an alarm signal in this regard. ~

Reviewer 2 Report

Comments and Suggestions for Authors

First of all, I would like to thank you very much for choosing our journal for your article. It is a very successful and meticulously prepared article. If you answer the questions I have asked, I would like to read the article again.

- Given the 10-day acclimatization period in 10 L aquariums, how did you ensure that the stress from environmental changes did not affect the baseline behavior and physiological state of the zebrafish before the experiment began?

- Could you clarify if there were separate control groups for each treatment (PE, PP, VPA, PEV, PPV) to account for potential synergistic effects, and how were these control groups maintained?

- How was the EthoVisionXT 14 software calibrated to ensure accurate tracking of zebrafish behavior, particularly for subtle behaviors like circling or inactivity?

- Was any correction method applied for multiple comparisons in the statistical analysis to control for the false discovery rate, given the multiple behavioral and oxidative stress parameters measured?

- How do you interpret the increase in specific SOD activity and the variable results for MDA levels in relation to oxidative stress and the overall health of the zebrafish? Can these results be directly linked to the behavioral changes observed?

- Considering the concentrations of pollutants used and the observed effects on zebrafish, how do these findings translate to real-world scenarios, particularly in aquatic environments where such pollutants are present?

- In order to increase the depth of your work, please also mention the recently popular topic of green composit and be sure to include the following two studies as references.
Physico-Mechanical Property Evaluation and Morphology Study of Moisture-Treated Hemp-Banana Natural-Fibre-Reinforced Green Composites
https://doi.org/10.3390/jcs7070266

Green composites: A review of processing technologies and recent applications
https://doi.org/10.1177/0892705718816354

- Given the high genetic similarity of zebrafish to humans and their absorption capacity for compounds, how do you justify the use of adult zebrafish for studying ASD-like behaviors and oxidative stress, considering the developmental stages at which VPA exposure is known to induce ASD-like symptoms?

- How were control groups designed to account for the potential synergistic effects of VPA and microplastics? Were there any vehicle controls or untreated groups included to distinguish the effects of the substances from the stress of handling and experimental procedures?

- Given the complex experimental design involving multiple treatments and behavioral assays, how was the sample size determined to ensure adequate statistical power to detect significant differences among treatments?

- How do the observed changes in SOD, GPx, and MDA levels contribute to our understanding of the oxidative stress response in zebrafish exposed to VPA and microplastics? Can these results be directly linked to specific behavioral changes observed in the study?

- Given the genetic similarity of zebrafish to humans and their use as a model organism in this study, what implications do your findings have for understanding the potential human health impacts of combined exposure to pharmaceuticals and microplastics?

Author Response

First of all, I would like to thank you very much for choosing our journal for your article. It is a very successful and meticulously prepared article. If you answer the questions I have asked, I would like to read the article again.

~Dear Reviewer 2, we are very grateful for your appreciation. Below you will find the answers to each of your questions, we hope that the context is now much clearer and better highlighted. ~

- Given the 10-day acclimatization period in 10 L aquariums, how did you ensure that the stress from environmental changes did not affect the baseline behavior and physiological state of the zebrafish before the experiment began?

~A 10-day acclimatization period is a standard procedure at the level of good practice guides (EU Commission Recommendation 2007/526/EC- Guidelines for the accommodation and care of animals used for experimental and other scientific purposes) but also in specialized literature, in some articles this period is even shorter, but we wanted to be sure. Regarding the stress from the environment, it is much reduced, our laboratory being a specific one for zebrafish in which we carry out only one experiment at a time and in which only those who take care of it enter, this lab has no other use. Moreover, they are located further away from the disturbance possibilities from the university (e.g. noise, agitation, vibrations) to ensure a quiet and comfortable working environment with them, eliminating as many possible stressors as possible, as well as having the windows covered so we can create artificially their light:dark cycle as needed. All the conditions for the well-being of the animals being mandatory in our lab. ~

- Could you clarify if there were separate control groups for each treatment (PE, PP, VPA, PEV, PPV) to account for potential synergistic effects, and how were these control groups maintained?

~There was a control group, which did not receive any kind of treatment to make sure that nothing interferes with the treatments, and the difference between sole PE and PP and the synergistic effect is given by the difference with the respective PE and PP after we administered VPA, through the continuous monitoring of those fish, as a timeline. ~

- How was the EthoVisionXT 14 software calibrated to ensure accurate tracking of zebrafish behavior, particularly for subtle behaviors like circling or inactivity?

~The calibration was a standard procedure carried out together with the product expert from NOLDUS. He specializes in programming the software to determine these subtle behaviors, that's what they deal with, most likely through AI, but here only those from Noldus can give a more pertinent answer about how they programmed the software. ~

- Was any correction method applied for multiple comparisons in the statistical analysis to control for the false discovery rate, given the multiple behavioral and oxidative stress parameters measured?

~There was no need to apply a data correction method in this work. In the case we apply correction methods, they would be mentioned. ~

- How do you interpret the increase in specific SOD activity and the variable results for MDA levels in relation to oxidative stress and the overall health of the zebrafish? Can these results be directly linked to the behavioral changes observed?

~An increase in superoxide dismutase (SOD) levels is generally a positive response to oxidative stress. SOD is an antioxidant enzyme that plays a crucial role in neutralizing superoxide radicals. Elevated SOD levels suggest an adaptive mechanism to counteract increased oxidative stress and protect cells from potential damage. MDA is the first and easiest to observe marker for lipid peroxidation, that is with cell membranes for example, as from the higher level of MDA we understand how great the effect of oxidative stress on cellular and tissue structures is, even more brain is formed in large part of lipids, and the biggest consumer of oxygen, but also the biggest producer of reactive oxygen species. In the matter of a possible link between oxidative stress and behavior, our research group showed in previous studies that there are correlations between oxidative stress and behavior (Cojocariu, R. O., Balmus, I. M., ... Ciobica, A., ... & Jurcoane, S. (2020). Camelina sativa methanolic and ethanolic extract potential in alleviating oxidative stress, memory deficits, and affective impairments in stress exposure-based irritable bowel syndrome mouse models. Oxidative Medicine and Cellular Longevity. https://doi.org/10.1155/2020/9510305). At present, our study did not have this practical correlation between behavior and oxidative stress as a key objective, but we are considering it for the future. ~

- Considering the concentrations of pollutants used and the observed effects on zebrafish, how do these findings translate to real-world scenarios, particularly in aquatic environments where such pollutants are present?

~Discussions in this sense can go in two directions if we refer to possible real-world scenarios: 1. the influence on fish in general and environmental pollution, changing the behavior of fish can lead to the extinction of the species, by changing the reproductive behavior, such as the increase/decrease in aggressiveness, decrease in sociability, decrease in reproduction behavior and rates; 2. human health – the presence of pharmaceutical residues in water, or from other sources, from which there is a possible cumulative effect affecting the health in the long term; the presence of microplastics in water and food sources, in the soil, etc., its arrival in the body and its accumulation, subsequently in the long term resulting in effects especially at the biochemical level through the evidence of oxidative stress, possible manifestation of neurodegenerative diseases or other neuropsychiatric implications can lead by the time also. ~

- In order to increase the depth of your work, please also mention the recently popular topic of green composit and be sure to include the following two studies as references.

Physico-Mechanical Property Evaluation and Morphology Study of Moisture-Treated Hemp-Banana Natural-Fibre-Reinforced Green Composites, https://doi.org/10.3390/jcs7070266

Green composites: A review of processing technologies and recent applications, https://doi.org/10.1177/0892705718816354

~Thank you for your recommendations. The articles are indeed very interesting; however, we could not quite find a strong connection of it to out article, so we decided to not include them at the current time. We will keep it in mind for other future articles. ~

- Given the high genetic similarity of zebrafish to humans and their absorption capacity for compounds, how do you justify the use of adult zebrafish for studying ASD-like behaviors and oxidative stress, considering the developmental stages at which VPA exposure is known to induce ASD-like symptoms?

~Zebrafish have a high degree of genetic similarity to humans, particularly in genes related to neurodevelopmental and behavioral processes. Testing on adult zebrafish is relevant in terms of developmental stage, as they have a fully developed brain. This allows researchers to examine the effects of acute or chronic exposure in an adult context, providing insights into behavioral and molecular changes associated with ASD-like symptoms and oxidative stress in mature organisms. There are several validated methods for ASD and zebrafish, more or less different, but this was not the purpose of our study, but only to highlight a comparison, regarding the use of VPA and for this aspect. Moreover, what we wanted to point out with this comparison is the very wide use of valproic acid both in the pharmaceutical and research area. In the present study, the aim was to evaluate the individual effects of polyethylene, polypropylene and valproic acid at environmentally relevant concentrations and their combined effects on the adult stage of Danio rerio at behavioral and biochemical levels, thus generating new information on the interactions between these pollutants and how they can generate antagonistic or synergistic effects. ~

- How were control groups designed to account for the potential synergistic effects of VPA and microplastics? Were there any vehicle controls or untreated groups included to distinguish the effects of the substances from the stress of handling and experimental procedures?

~The control group, did not receive any kind of treatment during experiments, but they were handled in the same way, for example when the water was changed, they participated in the same behavior tests, to make sure that nothing interferes with the administrated compounds, and the difference between sole PE and PP and the synergistic effect is given by the difference with the respective PE and PP after we administered VPA, through the continuous monitoring of those fish, as a timeline. ~

- Given the complex experimental design involving multiple treatments and behavioral assays, how was the sample size determined to ensure adequate statistical power to detect significant differences among treatments?

~First of all, we aim to respect the 3R rule as much as possible, minimizing the number of animals being essential, the choice of n=5 for each individual treatment is also based on other studies in the field. ~

- How do the observed changes in SOD, GPx, and MDA levels contribute to our understanding of the oxidative stress response in zebrafish exposed to VPA and microplastics? Can these results be directly linked to specific behavioral changes observed in the study?

~SOD, GPx, and MDA levels are the main biomarker studies in the literature regarding oxidative stress along with others such as CAT or ROS. Specifically, GPx is important because it is the enzyme in the defense apparatus, because it gets rid of oxygenated water, there are several sources of oxygenated water when oxidative stress occurs; GPx also has a related activity with H2O2, because it also neutralizes lipid peroxides, and cell membranes are mostly lipids, which helps us to see how it fights at the cellular level against cellular apoptosis via oxidative stress. MDA is the first and easiest to observe marker for lipid peroxidation, that is with cell membranes for example, as from the higher level of MDA we understand how great the effect of oxidative stress on cellular and tissue structures is, even more brain is formed in large part of lipids, and the biggest consumer of oxygen, but also the biggest producer of reactive oxygen species. Regarding the existence of a possible correlation between oxidative stress and behavior, our research group showed in previous studies that there are correlations between oxidative stress in the brain and behavior (Cojocariu, R. O., Balmus, I. M., ... Ciobica, A., ... & Jurcoane, S. (2020). Camelina sativa methanolic and ethanolic extract potential in alleviating oxidative stress, memory deficits, and affective impairments in stress exposure-based irritable bowel syndrome mouse models. Oxidative Medicine and Cellular Longevity. https://doi.org/10.1155/2020/9510305). At the present moment, our study did not have this practical correlation between behavior and oxidative stress as a key objective, but we are considering it for the future. ~

- Given the genetic similarity of zebrafish to humans and their use as a model organism in this study, what implications do your findings have for understanding the potential human health impacts of combined exposure to pharmaceuticals and microplastics?

~Exposure to these materials even in the case of humans can potentially have the same effects, probably better highlighted in a biochemical context, although human exposure to these contaminants is long-term. Even if we also consider some of their excretion capacity, visible effects will be long-term in the case of humans, not immediate effects, but we can especially expect biochemical imbalances in the first instance. ~

Reviewer 3 Report

Comments and Suggestions for Authors

This study performed an exploration of the toxicological impacts arising from the combination of microplastics and valproic acid on zebrafish. The interest in this research is particularly timely, given the increasing reports of pharmaceutical pollution. It would be highly beneficial to publish these findings promptly, following a few minor adjustments.

1) On line 63, it would be advantageous to include additional relevant studies to more clearly show the knowledge gap. By doing so, the significance of your work will be more prominently highlighted.

2) For lines 86-87, could you please incorporate more references to substantiate the choice of concentrations? It's important to demonstrate that these concentrations are reflective of those found in natural environments.

3) In the discussion section, kindly consider adding a paragraph to explore how the density and shape of microplastics may influence the outcomes of your study. For example, microplastics with a density greater than water tend to settle into the sediment, potentially reducing exposure time compared to microplastics of lower density. Additionally, the unique challenge posed by fibrous microplastics, which may entangle in the zebrafish's body, should be considered. In this context, the recent publication by Xiao, S., Cui, Y., Brahney, J., Mahowald, N. M., & Li, Q. (2023). Long-distance atmospheric transport of microplastic fibres influenced by their shapes. Nature Geoscience, 16(10), 863-870., could provide valuable insights.

4) Your findings on how different microplastics have varied responses in zebrafish, such as anxious behavior from PE and hyperactivity from PP, are intriguing. Could you please elaborate on the potential reasons behind these differing effects?

Author Response

This study performed an exploration of the toxicological impacts arising from the combination of microplastics and valproic acid on zebrafish. The interest in this research is particularly timely, given the increasing reports of pharmaceutical pollution. It would be highly beneficial to publish these findings promptly, following a few minor adjustments.

~Dear Reviewer 3, thank you for the all the kind words said about our article and for the suggestions you offered. We have specifically answered these below. ~

1) On line 63, it would be advantageous to include additional relevant studies to show the knowledge gap more clearly. By doing so, the significance of your work will be more prominently highlighted.

~We added additional relevant studies to the line you mentioned and rewrote the intent of our study, hopefully making it clearer (Line 68-75: From a pharmacological point of view, considering the beneficial effects, the studies presented the actions of VPA, such as promoting innovative neuroprotective, even antidiabetic and cardioprotective [26]. However, in addition to the effects in case of withdrawal of this drug, these also come with its potential side effects, such as hepatotoxicity, anemia, coagulopathy and teratogenic effects [27]. From a toxicological point of view, we are interested in clarifying the possible toxic effects that VPA can have as a result of unconscious exposure, especially from residues left in the water, at minimal doses. But also, whether or not microplastics can act synergistically with it.).

2) For lines 86-87, could you please incorporate more references to substantiate the choice of concentrations? It's important to demonstrate that these concentrations are reflective of those found in natural environments.

~The concentration was chosen considering the concentrations found in the environment. We also listed some of them in the introduction, but we also referred to them in the lines you mentioned. Other environmentally relevant tests that have been done on this compound in the literature, as opposed to animal models, were another reason for choosing this concentration. ~

3) In the discussion section, kindly consider adding a paragraph to explore how the density and shape of microplastics may influence the outcomes of your study. For example, microplastics with a density greater than water tend to settle into the sediment, potentially reducing exposure time compared to microplastics of lower density. Additionally, the unique challenge posed by fibrous microplastics, which may entangle in the zebrafish's body, should be considered. In this context, the recent publication by Xiao, S., Cui, Y., Brahney, J., Mahowald, N. M., & Li, Q. (2023). Long-distance atmospheric transport of microplastic fibres influenced by their shapes. Nature Geoscience, 16(10), 863-870., could provide valuable insights.

~We're grateful for the suggestion and have added it discussions section (Line 393-399: One aspect that needs to be considered is the shape of the microplastics used for the study and their capacity for transport. A study published in 2023 suggests that fibers are more effective at transporting over long distances than other shapes [58]. In our study, we have used both fibers as well as irregular flat shapes, but specific studies on this aspect should be taken into consideration in the future, along with an-other different analyses such as hormones or histology aspects to fully understand the mechanism of action of all the compounds of these polymeric materials and pharmaceuticals.) ~

4) Your findings on how different microplastics have varied responses in zebrafish, such as anxious behavior from PE and hyperactivity from PP, are intriguing. Could you please elaborate on the potential reasons behind these differing effects?

~Certainly, in order to identify solid reasons, more studies are needed regarding this aspect. The opinion we support in relation to this difference, especially at the level of anxiety, is the fact that the chemical composition of the polymers is different, so the more likely one of the compounds can be a specialized trigger for a certain type of behavior. Of course, further studies are necessary and vital to fully understand the mechanism of action of all the compounds of these polymeric materials. We hope that our study will open the door to these studies and encourage as many researchers as possible to join in identifying that specialized compound. ~

Reviewer 4 Report

Comments and Suggestions for Authors

Dear [Author/Authors],

I have had the opportunity to carefully review your manuscript titled "Toxicological response of zebrafish exposed to cocktails of pol- 2 ymeric materials and valproic acid." Overall, I find your work to be commendable and addressing a highly significant subject in the field.

Your data collection appears to be meticulously conducted, and the experimental methods are well-designed, contributing to the strength of your study. I have one minor comment that I believe could enhance the comprehensiveness of your research.

I would like to suggest exploring the internal morphology of the studied animals in more detail. Specifically, have you considered investigating additional physiological factors or health indices of the fishes? For instance, a histological analysis of the muscles or neurons could provide valuable insights into the overall physiological impact of environmental pollution on zebrafish.

If such analyses were not conducted, it would be beneficial if you could offer an explanation in the discussion or introduction as to why you chose not to investigate physiological issues. Clarifying the rationale behind this decision would contribute to a more comprehensive understanding of the study's scope and limitations.

Author Response

Dear [Author/Authors],

I have had the opportunity to carefully review your manuscript titled "Toxicological response of zebrafish exposed to cocktails of polymeric materials and valproic acid." Overall, I find your work to be commendable and addressing a highly significant subject in the field.

Your data collection appears to be meticulously conducted, and the experimental methods are well-designed, contributing to the strength of your study. I have one minor comment that I believe could enhance the comprehensiveness of your research.

I would like to suggest exploring the internal morphology of the studied animals in more detail. Specifically, have you considered investigating additional physiological factors or health indices of the fishes? For instance, a histological analysis of the muscles or neurons could provide valuable insights into the overall physiological impact of environmental pollution on zebrafish.

If such analyses were not conducted, it would be beneficial if you could offer an explanation in the discussion or introduction as to why you chose not to investigate physiological issues. Clarifying the rationale behind this decision would contribute to a more comprehensive understanding of the study's scope and limitations.

~Dear Reviewer 4, thank you for your kind words about our consistent work. We also consider that this histological investigation is very beneficial, adding extraordinary value to the understanding of the mechanisms of action of these pollutants, especially considering the fact that these microplastics can also generate physical effects in the body of fish, such as wounds of the digestive tract. Within our group, we were working with a histology specialist, but this being the first study carried out in combination, we wanted to see the first effects, the ones that are closest to us and on which we already have more experience, and for future studies, of course, we will go further with different finding including and considering histological/histopathological/IHC analysis. We mentioned this aspect now also in the discussion section and highlighted the need for further studies. ~

Reviewer 5 Report

Comments and Suggestions for Authors

I've read the manuscript (sustainability-2879014) in detail and my comments are as follows: 

General Comments:

In the manuscript, authors investigate the single and collective effects of common microplastics (polyethylene and  polypropylene) and pharmaceutical pollutants (valproic acid). Zebrafish is used as a test subject to examine the behavioral and biochemical effects of the above pollutants. The subject of the research is quite important in terms of understanding the effects of such common pollutants in the environment. I believe that the research contributes to the current literature positively. The research design is appropriate, and the organization of the paper is good. The introduction section provides sufficient background information from the literature. In my humble opinion, the quality of paper may be increased further if the points below are considered/corrected.      

Specific Comments: 

-Line 42-43: "Therefore, recent research has focused on the elimination of these contaminants [6]" What is the purpose of the citation at the end of the sentence?

-Line 56-58: Please check the meaning of the sentence. The sentence structure is not good and it is too long.

-Line 66: What do you mean by emphasizing "adult zebrafish". How does it affect the outcomes of the study? Can you give the average age of the fishes in terms of the reproducibility of your work in future studies?

-Line 80: "water changed daily". Is it possible to report the analysis result of the water that changed daily? What kind of water? Have you analyzed the water in terms of the bio-chemical composition before you add to the aquariums?

-Line 152: Long form of SEM.

-Line 256-263: I think this part should be moved to the section 2.2 or introduction.

Author Response

(The authors gave the same response as above.)

Round 2

Reviewer 2 Report

Comments and Suggestions for Authors

Accept in present form.